# A smooth tubercle bacillus from Ethiopia phylogenetically close to the *Mycobacterium tuberculosis* complex

Bazezew Yenew [1,6], Arash Ghodousi [2,3,6] ✉, Getu Diriba[1], Ephrem Tesfaye[1], Andrea Maurizio Cabibbe [3], Misikir Amare[1], Shewki Moga [1], Ayinalem Alemu [1], Binyam Dagne [1], Waganeh Sinshaw [1], Hilina Mollalign [1], Abyot Meaza [1], Mengistu Tadesse[1], Dinka Fikadu Gamtesa[1], Yeshiwork Abebaw[1], Getachew Seid[1], Betselot Zerihun [1], Melak Getu [1], Matteo Chiacchiaretta[3], Cyril Gaudin[4], Michael Marceau[4], Xavier Didelot [5], Getachew Tolera[1], Saro Abdella[1], Abebaw Kebede[1], Muluwork Getahun [1], Zemedu Mehammed[1], Philip Supply [4,7] ✉ & Daniela Maria Cirillo [2,3,7] ✉

The *Mycobacterium tuberculosis* complex (MTBC) includes several human- and animal-adapted pathogens. It is thought to have originated in East Africa from a recombinogenic *Mycobacterium canettii*-like ancestral pool. Here, we describe the discovery of a clinical tuberculosis strain isolated in Ethiopia that shares archetypal phenotypic and genomic features of *M. canettii* strains, but represents a phylogenetic branch much closer to the MTBC clade than to the *M. canettii* strains. Analysis of genomic traces of horizontal gene transfer in this isolate and previously identified *M. canettii* strains indicates a persistent albeit decreased recombinogenic lifestyle near the emergence of the MTBC. Our findings support that the MTBC emergence from its putative free-living *M. canettii*-like progenitor is evolutionarily very recent, and suggest the existence of a continuum of further extant derivatives from ancestral stages, close to the root of the MTBC, along the Great Rift Valley.

Tuberculosis (TB), a disease caused by *Mycobacterium tuberculosis* and other members of *the M. tuberculosis* complex (MTBC), has been a major killer throughout human history[1,2]. Based on the current knowledge, the MTBC is comprised of nine human-adapted lineages (L), representing *M. tuberculosis sensu stricto* (L1-4 and L7-8) and *M. africanum* (L5-6 and L9), plus at least nine animal-adapted lineages[3–5]. Strains belonging to these lineages share highly conserved genomes, differing by no more than ~2000 single nucleotide polymorphisms (SNPs), and with no significant trace of genetic exchange among them, suggesting that this entire population expanded clonally from a single

bacterial progenitor[6–8]. As seen in host adaptation of other bacterial pathogens[9], this single-cell bottleneck might reflect a drastic change in ecology and selective pressures that putatively coincided with the jump to an obligate pathogen lifestyle in a mammalian host[10].

In accordance with this scenario, the current view is that the MTBC progenitor clone emerged from a diversified and recombinogenic pool of free-living mycobacteria resembling extant strains of *M. canettii*, which is the phylogenetically closest relative of the MTBC[11–13]. In contrast to the worldwide distribution of the MTBC, clinical isolates of *M. canettii* are geographically restricted to the Horn of Africa, and

[1]Ethiopian Public Health Institute, Addis Ababa, Ethiopia. [2]Vita-Salute San Raffaele University, Milan, Italy. [3]IRCCS San Raffaele Scientific Institute, Milan, Italy. [4]Univ. Lille, CNRS, Inserm, CHU Lille, Institut Pasteur de Lille, U1019 - UMR 9017 - CIIL - Center for Infection and Immunity of Lille, F-59000 Lille, France. [5]School of Life Sciences and Department of Statistics, University of Warwick, CV4 7AL Coventry, UK. [6]These authors contributed equally: Bazezew Yenew, Arash Ghodousi. [7]These authors jointly supervised this work: Philip Supply, Daniela Maria Cirillo. ✉e-mail: ghodousi.arash@hsr.it; philip.supply@ibl.cnrs.fr; cirillo.daniela@hsr.it

fewer than 150 isolates have been described, mainly obtained from Djibouti residents. There is no evidence of inter-human transmission of *M. canettii*[13,14], and *M. canettii* strains are less persistent and virulent than MTBC strains in infection models[12,15], indicating a lower degree of pathoadaptation. Moreover, in contrast with MTBC members, *M. canettii* strains show marked genome mosaicism indicating multiple intraspecies horizontal gene transfer (HGT) events, explained by a distributive conjugal transfer (DCT) mechanism and consistent with frequent inter-strain contacts in an environmental reservoir[11,12,16]. Furthermore and consistently with the hypothesized *M. canettii*-like progenitor of the MTBC, recent phylogenomic analyses suggested that HGT events have also happened with ancestors of extant *M. canettii* strains in the ancestral evolutionary branch leading to the MTBC genesis. This finding provides additional support for a shared gene pool and ecological niche between both parental populations[17].

This hypothesis is further supported by the observation of homologous recombination spontaneously occurring in vitro between two adjacent *pks5* genes in *M. canettii*, resulting in abrogation of lipooligosaccharides (LOS) production and a change in colony morphotype from smooth to rough. This recombination results in a single *pks5* configuration similar to that conserved among all MTBC members, which lack LOS and show a rough morphotype. This conservation suggests a unique recombination event in the MTBC progenitor after separation from its *M. canettii*-like ancestor[18]. Importantly, such *M. canettii* morphotype variants show increased virulence in cellular- and animal-infection models, suggesting that *pks5* recombination generated a beneficial genetic background in the MTBC ancestor for colonization and spread in mammalian hosts[18].

However, when this emergence from the recombinogenic *M. canettii* pool and these key ecological and adaptive changes happened in the ancestral branch of the MTBC remains unknown. Indeed, previous analysis showed that *M. canettii* strains can differ by up to 65,000 SNPs from each other, and at least 14,000 SNPs separate the hitherto known closest lineages from the currently acknowledged most recent common ancestor (MRCA) of the MTBC[12].

Here, we report the discovery of a TB clinical isolate (hereafter referred to as ET1291) from Ethiopia, which shares the characteristic smooth colony morphology and twin-*pks5* configuration of *M. canettii* strains, but is phylogenetically much closer to the MTBC clade. We used short and long read sequencing to reconstruct the complete circular genome of the ET1291 strain and performed in-depth comparative genomics with all previously available and other newly sequenced *M. canettii* genomes and representatives of all known MTBC lineages. The obtained findings provide proximal insights into the genomic and biological changes involved in the emergence of the MTBC, and on its original lifestyle.

## Results

### Patient with the ET1291 infection in Ethiopia
ET1291 was isolated from a newly diagnosed 20-year-old HIV-negative male patient living in the rural part of central Tigray, Ethiopia. The strain was identified within the national anti-TB drug resistance survey 2017–2019, designed as a cross-sectional health facility-based study as per WHO recommendations[19], enrolling for 16 months newly registered and previously treated bacteriologically confirmed pulmonary TB cases. The primary culture of ET1291 became positive after 6 days using BD BACTEC MGIT liquid culture and colonies were observed after 5 weeks on the Löwenstein–Jensen medium. Such colonies showed the archetypal smooth morphotype that distinguishes *M. canettii* strains from MTBC strains. Extended drug susceptibility testing showed monoresistance to pyrazinamide, consistent with the natural resistance of *M. canettii* to this drug[20]. Moreover, ET1291 showed a faster growth rate in liquid medium (Middlebrook 7H9 supplemented with OADC and glycerol) than *M. tuberculosis* H37Rv, similarly to a reference *M. canettii* strain STB-A (CIPT

140010059) (Supplementary Fig. 1) and as also seen with other *M. canettii* strains[12].

### WGS analysis, genotypic resistance, SNP profile, and phylogenetic position of ET1291
Analysis with the MTBseq pipeline[21] of Illumina-based whole-genome sequencing (WGS) data obtained from ET1291 (Fig. 1) was unable to provide a *M. canettii* or MTBC phylogenetic classification. Phylogenetic SNPs detected across the genome were shared with evolutionarily early branching MTBC lineages (L1, 5, 6, 7, 8 and animal lineages; to note, reference phylogenetic SNPs have not been defined for L9 yet) and/or *M. canettii* (Supplementary Data 1). Of note, multiple non-synonymous SNPs (17 and 2, respectively; Supplementary Data 2) were found in the *fgd1* and *Rv0678* genes, associated with resistance to the newest anti-TB drugs delamanid/pretomanid and bedaquiline, respectively; however, the minimum inhibitory concentrations of ET1291 for all three drugs were below the proposed respective critical concentrations. Targeted next-generation sequencing (tNGS) using Deeplex Myc-TB testing[22] resulted in *M. canettii* identification with a predicted pyrazinamide mono-resistant profile, based on *hsp65* sequencing data and detection of specific phylogenetic SNPs including *pncA* A46A (Fig. 1). Like for previous *M. canettii* strains[12,23], no spoligotype spacers were detected, indicating absence of spacers in common with those in the CRISPR-Cas locus of reference MTBC genomes, which was confirmed by analysis of the WGS data using CRISPRCasFinder (https://crisprcas.i2bc.paris-saclay.fr/). This *M. canettii* classification was supported by multiple other archetypal characteristics shared with previously described *M. canettii* strains, as described below.

To determine the phylogenetic position of ET1291, a maximum likelihood phylogeny was inferred from a whole-genome alignment of 80 MTBC genomes, including representatives of all known human- and animal-adapted lineages, as well as 39 known publicly available or newly sequenced *M. canettii* strains. This reference dataset of *M. canettii* strains represents the closest outgroup to ET1291 strain and the MTBC, as shown by in-depth analysis of the ET1291 genome (see below) and recent genomic analysis of phylogenetically closest non-tuberculous mycobacterial species, comprising *M. decipiens, M. shinjukuense, M. lacus,* and *M. riyadhense* (forming a clade defined as the "MTB-associated phylotype")[24] (Supplementary Data 3). A unique phylogenetic position was thereby discovered for ET1291, revealing an intermediate branch between the reconstructed common ancestral node of the MTBC lineages and the closest node of the other, currently known *M. canettii* strains (Fig. 2a). This median position was confirmed by a reconstructed phylogeny based on 1000 single copy core genes, additionally including strains of the MTB-associated phylotype as more external outgroups (Supplementary Fig. 2). Genome-wide SNP data based on 82,637 positions indicated that ET1291 was separated by 5948 SNPs from the reconstructed ancestral genome of the MTBC, and at least 8984 SNPs from the next closest, previously described *M. canettii* strain, indicating a substantially closer genetic distance of ET1291 relatively to the MTBC compared to *M. canettii* strains documented thus far.

### Estimated ratios of recombination versus mutation for ET1291, *M. canettii,* and MTBC
We then investigated whether ET1291 was part of the recombinogenic population of *M. canettii*, and sought to evaluate the impact of recombination on the genetic relationships identified above. To do so, we used ClonalFrameML, which also estimates a phylogeny accounting for recombination, thereby correcting for skewing effects due to potential imports of genomic segments[25]. This analysis detected multiple recombination segments in the core genome of ET1291 strain, as in previously identified *M. canettii* strains, corresponding to HGT events that occurred after the separation from these *M. canettii*

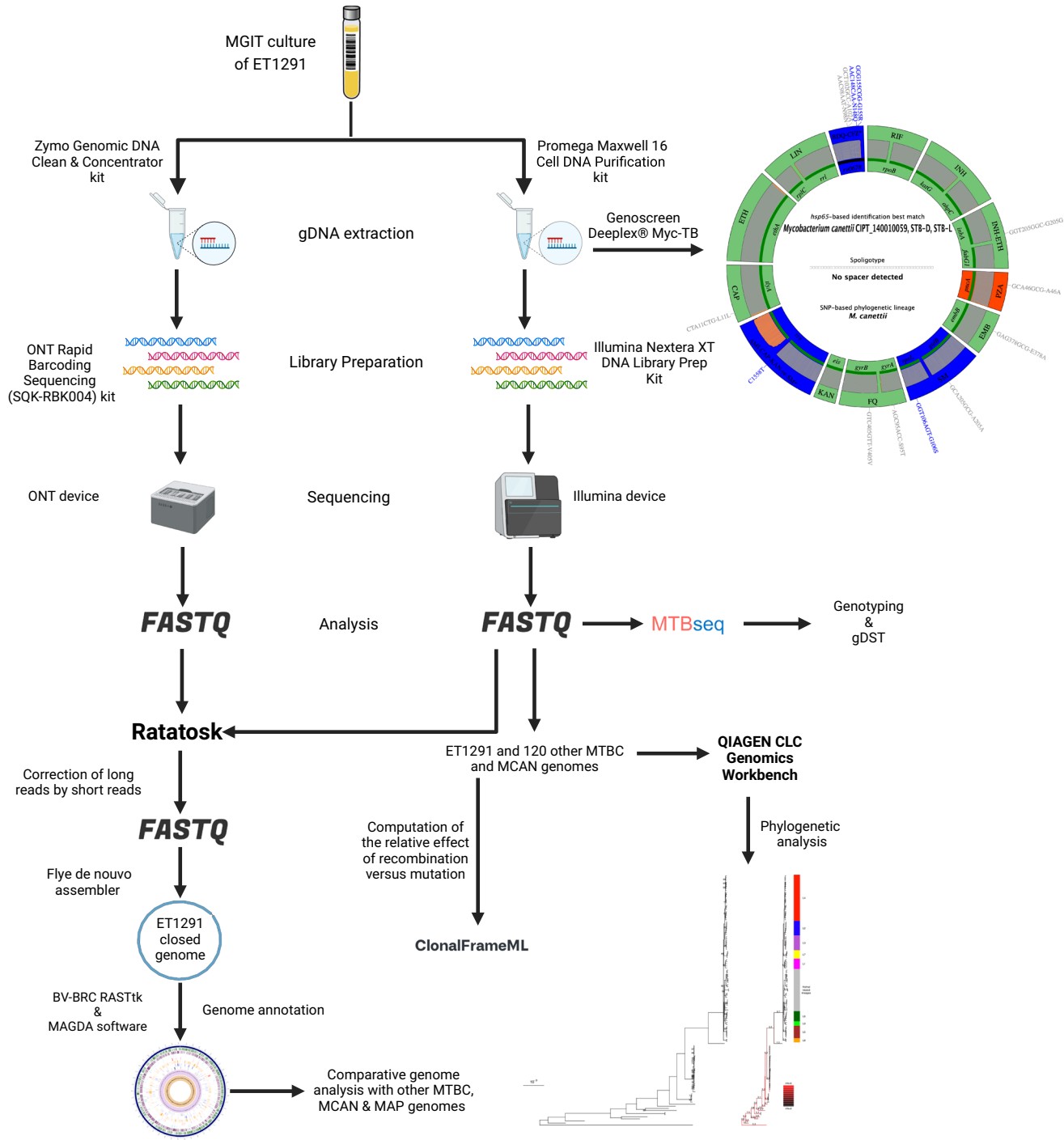

**Fig. 1 | Schematic workflow showing main steps of sequencing and post-sequencing analyses of the study.** The hybrid strategy combining Illumina and Oxford Nanopore sequencing used to obtain the complete assembly and functional annotation of the genome of *M. canettii* ET1291, and to perform comparative genomics analyses with previously known *M. canettii* and MTBC strains, are represented in the center. Illumina whole-genome sequencing data were also used to perform SNP-based phylogenetic reconstructions and to determine the ratio of recombination versus mutation in the strain dataset. Results of targeted next-generation sequencing using the Deeplex Myc-TB assay, identifying ET1291 as *M. canettii*, are represented on the circular map on the right. Information on *hsp65* best match-based and phylogenetic SNP-based identification of *M. canettii* is shown inside the circle, along with information on spoligotype (without any spacer detected). Target gene regions in the map are grouped within sectors according to the anti-tuberculous drug resistance with which they are associated. Sectors in red (for pyrazinamide (PZA) here, as typically expected for *M. canettii*) and green indicate targets in which resistance-associated mutations or either no mutation or only mutations not associated with resistance (shown in gray) are detected, respectively. Blue sectors refer to regions where as yet uncharacterized mutations are detected (see ref. 22 for further details on the Deeplex map). gDNA, genomic DNA; MTBC, *M. tuberculosis* complex, MCAN, *M. canettii*; MAP, *M. tuberculosis* (MTB)-associated phylotype. Abbreviations of anti-tuberculous drugs on the Deeplex map: RIF: rifampicin, INH: isoniazid, PZA: pyrazinamide, EMB: ethambutol, SM: streptomycin, FQ: Fluoroquinolones, KAN: kanamycin, AMI: amikacin, CAP: capreomycin, ETH: ethionamide, LIN: linezolid, BDQ: bedaquiline, CFZ: clofazimine. Figure created using BioRender.com by Arash Ghodousi with license to publish.

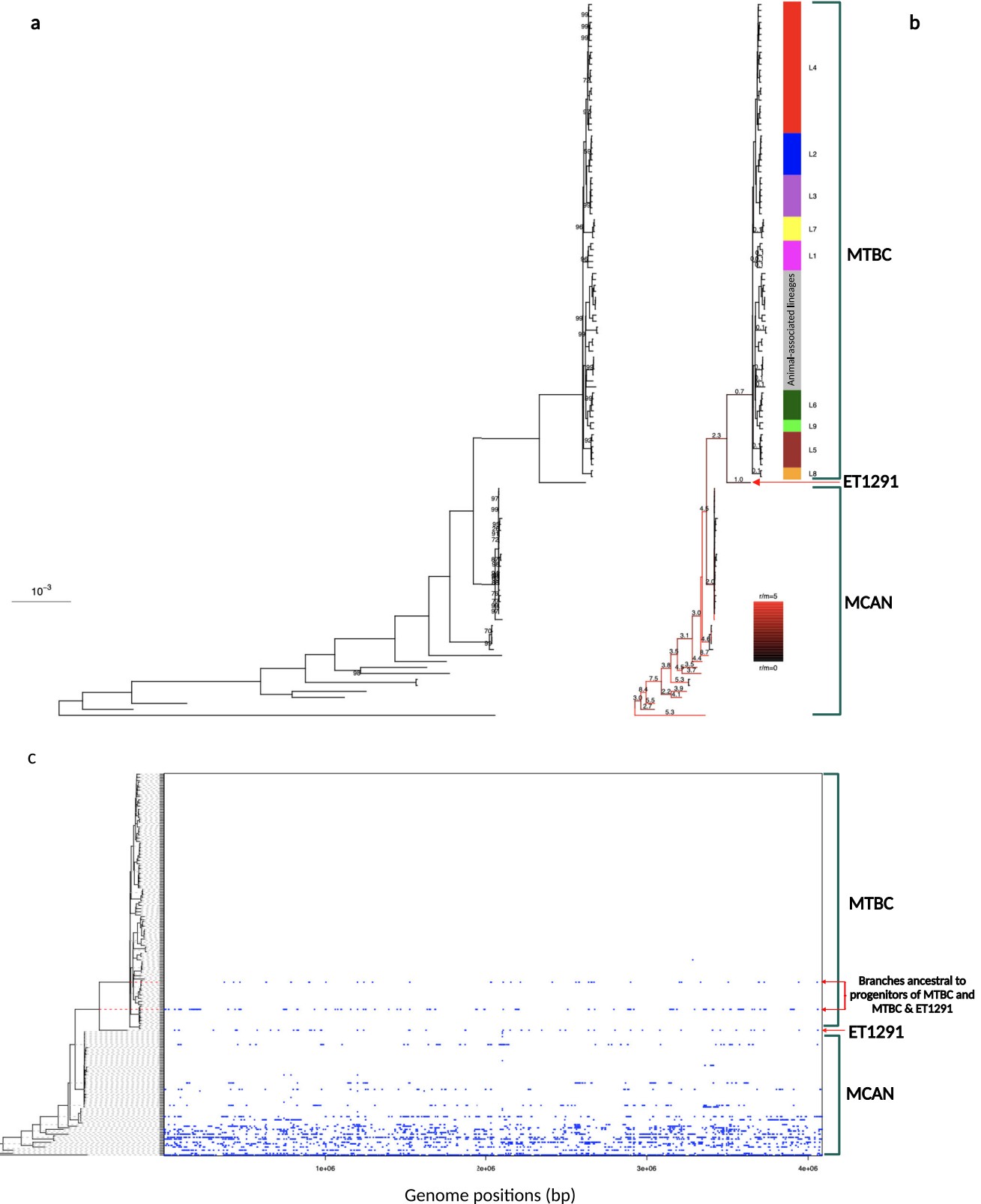

lineages (Fig. 2b, c). Moreover, subsequent HGT events were mapped on the branch leading to the MTBC ancestor, indicating continuation of gene exchanges even after its divergence from the ET1291 lineage. Of note, substantial variations were observed in the estimated ratios of recombination vs mutation (r/m) between different branches of the phylogeny (Fig. 2b, c). Strikingly, the r/m ratio fell to 1 or below (0.7–1.0) in the branches leading to ET1291 and the MTBC ancestor, in contrast to the main, earlier branching *M. canettii* lineages (2.0–8.7), thus indicating reduced lateral gene flow closer to the emergence of the MTBC progenitor (Fig. 2b). Consistently with current evidence supporting strict clonality[8,16,26], virtually no trace of recombination was detected in the MTBC. After correction for multiple recombination, lengths of the branches leading to *M. canettii* strains and the MTBC ancestor were considerably shorter in the ClonalFrameML tree

**Fig. 2 | Whole-genome alignment-based phylogeny and recombinogenic versus clonal structure of ET1291, other *M. canettii* and MTBC strains. a** Maximum likelihood phylogenetic tree including 121 genomes before accounting for recombination. Bootstrap support values are shown only for values below 100%; the value is not shown for branches with 100% bootstrap support, which represent the majority of the long branches as expected. **b** Maximum likelihood phylogenetic tree including the same genomes after correction for recombination. Branches are colored according to the estimated ratios of recombination versus mutation (r/m). r/m values are only shown for branches longer than 6e-5 substitutions/site, because of inherent stochasticity in signal to noise ratios in the shortest branches. The intermediate phylogenetic position of ET1291 between the MTBC and the other, currently known *M. canettii* strains is indicated by a red arrow. **c** ClonalFrameML analysis of the 121 genomes, showing the extent and segments of recombination mapped on the branches of ET1291 and other *M. canettii* strains and the branch leading to the common ancestor of the MTBC. Note that the red dotted lines indicate two branches that are ancestral to the progenitor of the MTBC and to the progenitor of the MTBC and ET1291, respectively, and are thus not part of the MTBC clade. MTBC, *M. tuberculosis* complex; MCAN, *M. canettii*. ET1291 indicated by a red arrow.

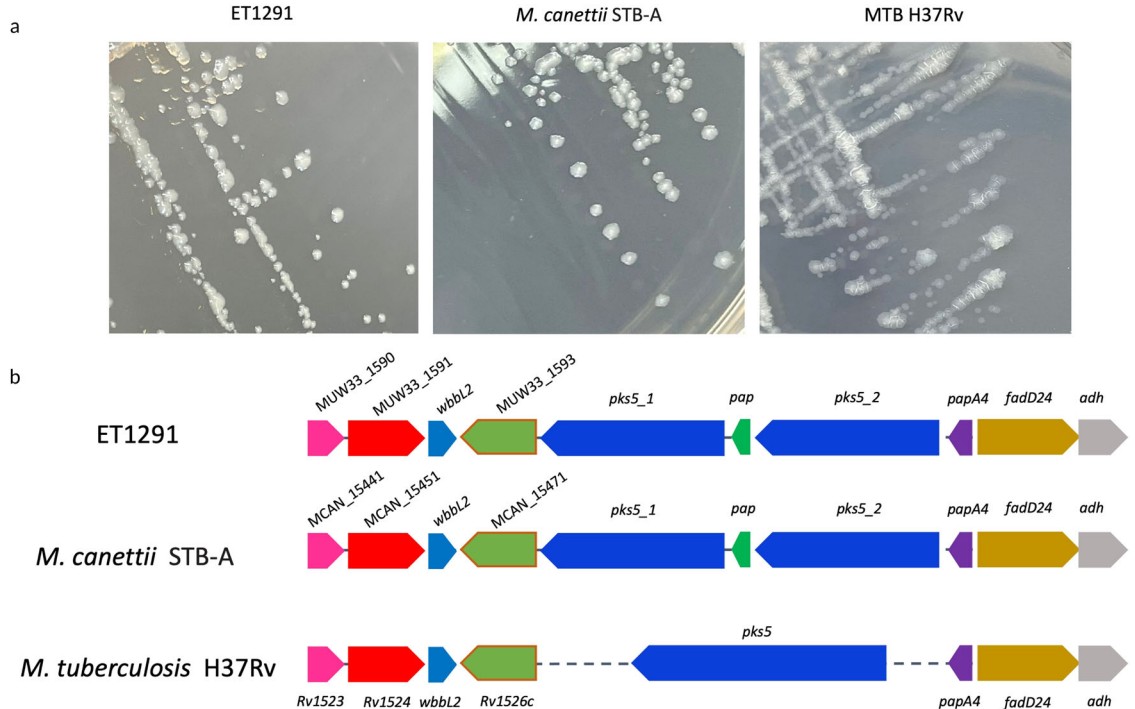

**Fig. 3 | Colony morphology and architecture of the *pks5* genome region in ET1291, *M. canettii* STB-A, *M. tuberculosis* H37Rv. a** Similar smooth colony morphotypes shared by ET1291 and *M. canettii* STB-A, contrasting with the rough colony morphotype of *M. tuberculosis* H37Rv. **b** Aligned genome segments showing the dual polyketide synthase-encoding *pks5* gene configuration with an intervening *pap* gene shared by ET1291 and *M. canettii* STB-A (and all other *M. canettii* strains), instead of the single *pks5* gene found in *M. tuberculosis* H37Rv (and all other MTBC members). Colors show the correspondence between gene orthologues among ET1291, *M. canettii* STB-A, and *M. tuberculosis* H37Rv.

(Fig. 2b). Based on the inferred clonal genealogy, the distance between ET1291 and the last common ancestor of the MTBC was reduced to 3389 SNPs, which was only two-fold larger than the maximal distance separating any two MTBC strains (1735 SNPs commonly covered in all studied genomes, Fig. 2b).

**Defining features of the complete ET1291 genome**

To comprehensively identify the genomic features associated with the proximal phylogenetic position of ET1291 relatively to the MTBC, the ET1291 strain was subjected to Oxford Nanopore Technologies (ONT) MinIon long read sequencing, followed by polishing and hybrid assembly using Illumina short reads. This hybrid approach resulted in a single complete circular genome of 4,515,631 bp containing 4086 predicted genes (Supplementary Fig. 3, Supplementary Data 4). This genome size is ~85–175 kb larger than the 4.34–4.43 Mb size range of the MTBC genomes (4.41 Mb for *M. tuberculosis* H37Rv), similarly to *M. canettii* genomes (currently culminating at 4.53 Mb for STB-K) previously identified[12]. As reductive evolution is a hallmark of mycobacterial pathogens[26,27], this larger genome size relative to the MTBC further supports an ancestral status shared with *M. canettii* strains.

Additional ancestral characters were found in common with known *M. canettii* genomes. Consistent with its typical smooth colony morphotype, ET1291, displayed the dual polyketidesynthase-encoding *pks5* gene configuration with an intervening pap gene shared by all *M. canettii* strains, instead of the single *pks5* gene found in all MTBC members (Fig. 3)[18]. Moreover, the orthologues of four interrupted coding sequences (ICDS) in the MTBC (Supplementary Data 5), reflecting frameshifts or stop codons acquired during pseudogenization of the MTBC genomes[28], were intact in ET1291, as in the formerly characterized *M. canettii* genomes and the non-tuberculous mycobacterial species outgroup[12]. Therefore, these frameshifts, and the recombination in the *pks5* locus that is thought to have resulted in surface remodeling and increased virulence in the MTBC MRCA[18], likely occurred only after the separation of the ancestral branch leading to the MTBC from the ET1291 lineage. Further like all *M. canettii* strains, ET1291 contains the *cobF* region, involved in the cobalamin (vitamin B12) synthesis, only shared by strains of the MTBC L8, representing a sister clade compared to the rest of the MTBC[4].

Nevertheless, other genome features are consistent with an evolutionarily closer branching point relatively to the MTBC. The average nucleotide identity versus *M. tuberculosis* H37Rv of the ET1291 genome

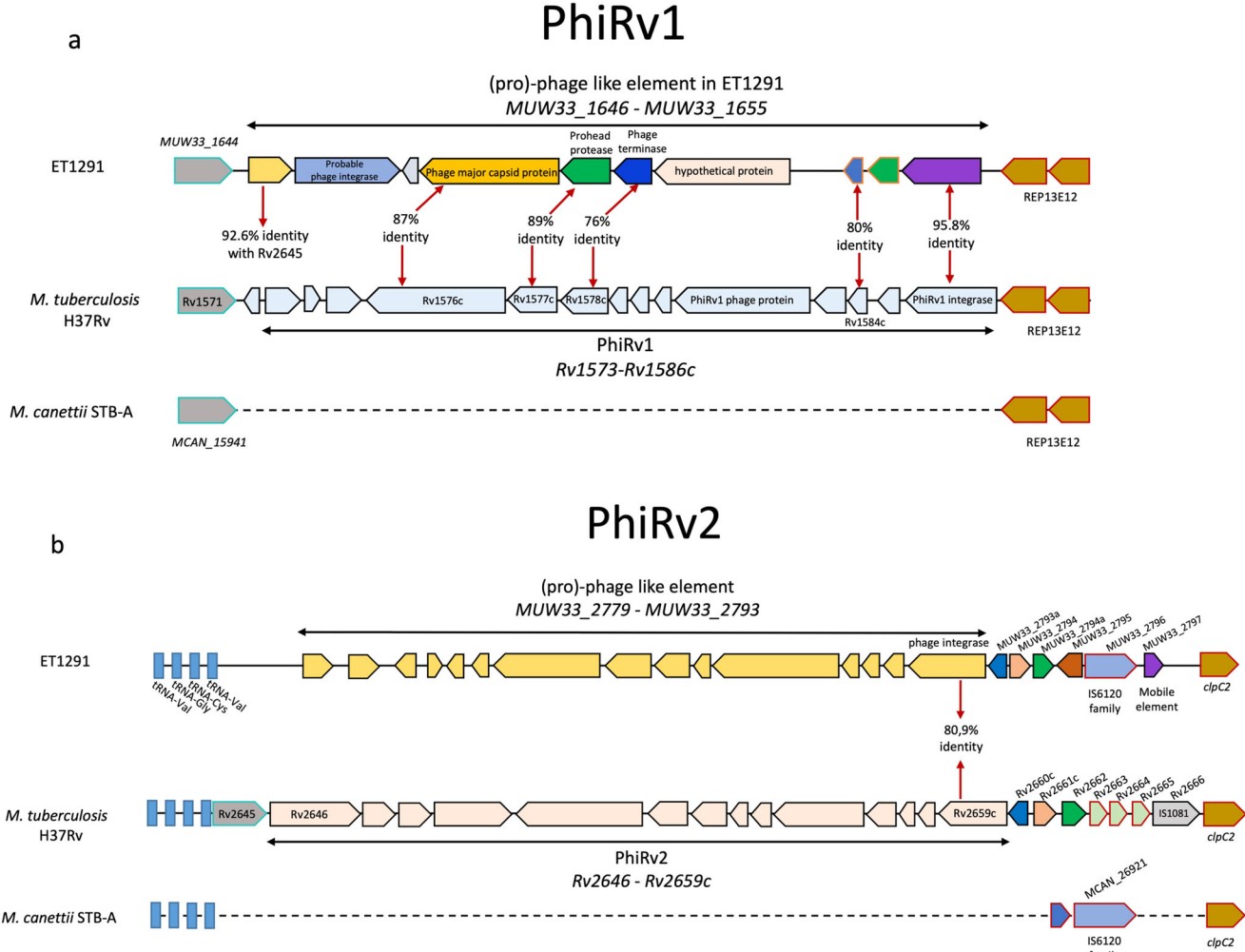

**Fig. 4 | Presence of two prophage-like regions in ET1291 unlike in other *M. canettii* strains. a** PhiRv1 prophage region. **b** PhiRv2 prophage region. These two prophage-like regions of ~9–10 kb found are located in the same genomic locations as the PhiRv1 and PhiRv2 prophage-like regions of MTBC strains. Percent identities are shown for genes with at least 70% identity with an orthologue in *M. tuberculosis* H37Rv. Only a fraction of these phage gene sequences show substantial identity (76–96%, lower than the 99–100% seen for the core genes) with the MTBC PhiRv1 and PhiRv2 genes, suggesting that these prophages were acquired independently from those in the MTBC.

is 99.55%, incrementally above the 97.97–99.37% range observed with all the previously described genomes of *M. canettii* strains[12] (Supplementary Data 6). Moreover, the ET1291 genome contains the orthologues of eight of the 50 MTBC genes that were not found in the previously identified *M. canettii* genomes[12] (Supplementary Data 7) and that partially overlap with genomic islands likely acquired by HGT in the MTBC[29]. Excluding the MTBC, the sequences of this gene subset show best hits with ~70–90% identity at the nucleotide level with counterparts in different environmental mycobacteria, including *M. shinjukuense* and *M. riyadhense* part of the MTB-associated phylotype, or mycobacteria part of the *M. avium* complex (Supplementary Data 8). Proteins encoded by the 8-gene set are a probable transcriptional regulatory protein (Rv1990c; the corresponding gene is essential for in vitro growth of *M. tuberculosis* H37Rv[30], a member of the PE_PGRS protein family (Rv0746c, PE_PGRS9), a probable fatty-acid-CoA synthetase (Rv3513c, fadD18), as well as proteins of unknown functions. The corresponding genes were thus probably acquired already by lateral transfer in a common ancestor of ET1291 and the MTBC, before the emergence of the MTBC MRCA.

Further in contrast to formerly known *M. canettii* strains, the ET1291 genome also contains prophage regions of ~9–10 kb found in the same genomic locations as the PhiRv1 and PhiRv2 prophage regions of MTBC strains (Fig. 4). However, only a fraction of these phage genes show substantial identity (83–96%, lower than the 99–100% seen for the core genes) with the MTBC PhiRv1 and PhiRv2 genes, suggesting that these prophages were acquired independently from those in the MTBC. Interestingly, a third, 7.1 kb prophage region was additionally found, located between the orthologues of *Rv1719-Rv1720c* of H37Rv (Supplementary Fig. 4). In addition to genes with 83–91% identity with those in the PhiRv1 and PhiRv2 prophages in H37Rv, other genes encoded by this region are 70–75% identical to those found in non-tuberculous mycobacteria such as the MTB-associated phylotype member *M. shinjukuense*, and *M. koreense*, both with an unknown reservoir (Supplementary Data 9).

As another prominent imprint of HGT, ET1291 contains a CRISPR-associated protein (CRISPR-Cas) locus encoding a system of type I-E, also present with 98.7-100% identical *cas* genes in *M. canettii* STB-G, STB-I[12] and Mc157 (ERR266126) (Fig. 5). However, the 115 spacers found in the CRISPR-Cas locus of ET1291 appear strain-specific. ET1291 is phylogenetically distant from *M. canettii* STB-G, STB-I, and Mc157 (Supplementary Fig. 5), indicating that the locus was horizontally exchanged among these strains or from a common donor, instead of having been inherited from a common ancestor shared by these four strains. The distribution of the other types of CRISPR-Cas systems

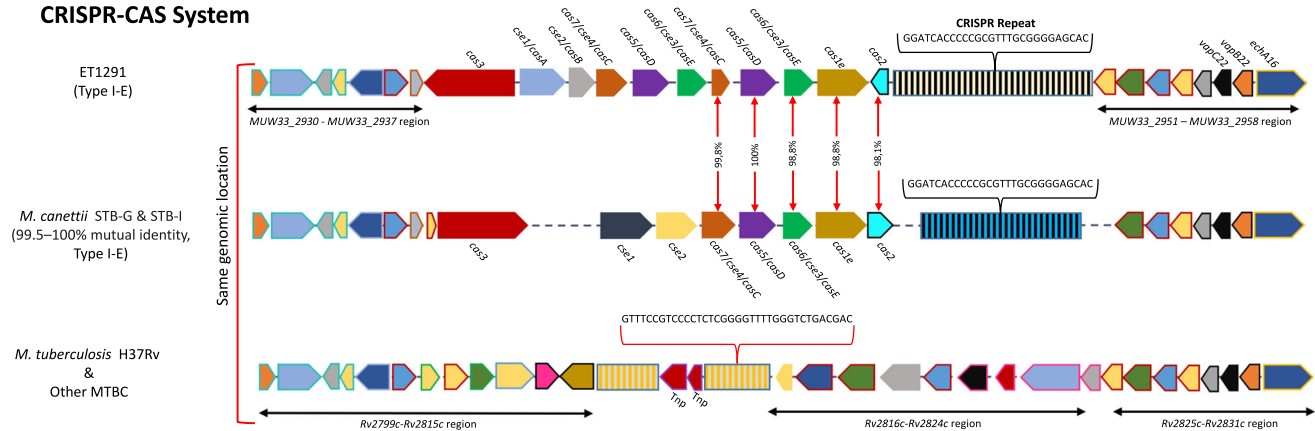

**Fig. 5 | CRISPR-Cas systems in ET1291, STB-G, STB-I, and *M. tuberculosis* H37Rv.** Percentages of identities are indicated with arrows between *cas2*, *cas1e*, *cas6/cse3/ casE*, *cas5/casD* and *cas7/cse4/casC* genes of ET1291, and STB-G/I. CRISPR repeats sequences (shown above the respective CRISPR arrays) are identical between ET1291, STB-G, and STB-I (as shown as same black vertical bars in CRISPR arrays), while CRISPR spacers are strain-specific (as shown by different colors interspaced between vertical bars among respective CRISPR arrays). Colors show the correspondence between gene orthologues among ET1291, STB strains, and *M. tuberculosis* H37Rv.

mapped on a phylogenetic tree (Supplementary Fig. 5) was similarly homoplasic, indicating multiple independent HGT-mediated acquisition events, including possibly for the system of type III-A acquired in the same genomic region in the ancestral branch of the MTBC.

HGT-mediated exchanges or acquisitions from a common ancestral gene pool between *M. canettii* and the MTB-associated phylotype were further denoted by the finding that genomic regions of the MTBC members such as *Rv0402c-Rv0406*, *Rv1041c-Rv1050*, *Rv1499–1510* and *Rv2307c-Rv2309c* were replaced in the genomes of ET1291 and/or *M. canettii* STB-J and -K by only distantly related or unrelated gene segments, with coding sequences sharing ~90–99% identity at nucleotide level with their equivalents in *M. shinjukuense* (or >80% identity with homologs in *M. riyadhense*) (Supplementary Figs. 6–8).

## Discussion

Tracing the origins and conditions of emergence of the MTBC provides fundamental clues to understand the causes of its outrageous evolutionary success[8,12,27,31–33]. Whether lineages of MTBC strains, now extinct, could have existed and caused TB in humans and animals much earlier (maybe by two orders of time magnitude) than the MRCA of known extant lineages, is an open and largely debated question[10,34,35]. The discovery in East Africa of a TB clinical isolate that shares typical features of *M. canettii* strains but is phylogenetically much closer to the MTBC reveals that this was likely not the case, indicating that the emergence of the ancestral clone of the MTBC from its putative free-living *M. canettii*-like progenitor pool is evolutionarily very recent.

After correction for recombination, the clonally inherited SNP distance separating ET1291 from the currently known MTBC MRCA is only about two-fold the maximal distance separating any two known extant MTBC strains. Consistent with this proximal phylogenetic position, while ET1291 shares archetypal phenotypic and genomic characteristics of *M. canettii*, including its smooth colony morphotype associated with the twin-*pks5* gene configuration, genome-wide imprints of HGT and a larger genome size compared to the MTBC, it distinctively harbors some specific genes likely acquired in a recent common ancestor shared with the MTBC MRCA. By hypothesizing similar molecular clocks across the respective evolutionary branches, this inferred relative phylogenetic distance would thus imply that the original MTBC progenitor that emerged from a *M. canettii*-like ancestor in East Africa is perhaps only twice as old evolutionarily than the currently reconstructed MRCA of the MTBC. More data would be

required for estimating a dating for this emergence, also in the view of the debated dating of the MTBC MRCA[32,35].

Regardless of temporal estimation, the multiple traces of HGT in the ET1291 genome and those inferred, after the separation of the ET1291 lineage, in the evolutionary branch leading to the MTBC represent evidence for continued exposure to lateral gene flow until the emergence of the MTBC progenitor clone. Consistent with previous presumption[17], these results imply at least partially shared habitats between ancestors of *M. canettii* strains, including those part of the ancestral branch of the MTBC, but also other environmental (myco)bacterial (including members of the MTB-associated phylotype) and/or phage populations. However, our data suggest an increased barrier to gene flow during evolution in the branches phylogenetically closest or leading to the MTBC, before becoming extreme within the MTBC[16,17], post-emergence. This raises intriguing questions of whether such increase already reflected partial adaptation to a new, potentially host-associated ecological niche separate from the rest of the ancestral free-living *M. canettii* strain pool, and/or from relative reduction of natural abilities of some ancestral strains to receive extrinsic DNA. Both hypotheses are consistent with certain experimental findings on previously known *M. canettii* strains. Indeed, some correlation was observed between phylogenomic distance of these strains from the MTBC and degrees of virulence/persistence in animal infection models[12]. Moreover, some, but not all, extant *M. canettii* strains can act in vitro as recipients of chromosomal segments transferred from MTBC or other *M. canettii* donor strains, while MTBC strains appear incapable to do so[36]. Further experiments would be needed to determine the degree of virulence/persistence of ET1291 in infection models and to which extent this strain has retained a recombinogenic property.

ET1291 was the sole *M. canettii*-like strain identified in the national drug resistance survey conducted in Ethiopia. Likewise, only two *M. canettii* strains were identified in another national survey recently conducted in the same geographic region[37], including one representing an additional intermediate lineage between the MTBC and the formerly identified *M. canettii* strains, albeit with a branching point more distant from the MTBC (and thus closer to the previously known *M. canettii* strain pool) compared to ET1291 (in preparation). *M. canettii* strains represented a more notable, but still small, proportion (i.e., 6%) of the TB isolates only in a national survey previously done in Djibouti[38]. These population-based proportions represent evidence of the infrequency of human TB caused by *M. canettii* even in the Horn of Africa, further arguing for spillover infection from a local

environmental reservoir. It is thus all the more fascinating that extant representatives of *M. canettii* lineages reflecting distinct ancestral stages towards the obligate MTBC pathogens can still be sporadically isolated from tuberculosis patients living in different places of this region. When also combined with the recent identification of a new sister clade of the MTBC in the African Great Lakes region[4], these findings suggest that a continuum of extant derivatives from further ancestral stages near the root of the MTBC may well exist along the Great Rift Valley. With wider use of genome sequencing in the region, more strains might thus be uncovered, which will help further dissect and resolve the different molecular and evolutionary events leading to the emergence of *M. tuberculosis*. Last but not least in this respect, we showed that *M. canettii* strains even genetically very distant from the MTBC have the potential to rapidly evolve towards enhanced, *M. tuberculosis*-like virulence and persistence phenotypes, via spontaneous *pks5* recombination in vitro[18] or by different mutational pathways during in vivo experimental evolution[33], respectively. Taken altogether, these data thus suggest a possibility of (re-)emergence of a (novel) prototype of professional TB pathogen.

## Methods
The ET1291 strain was isolated from a patient with TB in the framework of a national drug resistance survey in Ethiopia according to WHO recommendations.

### Strain characterization
The ET1291 strain was isolated from a patient with TB in the framework of a national drug resistance survey in Ethiopia (manuscript in preparation) according to WHO recommendations[19]. Phenotypic drug susceptibility testing (DST) for first-line drugs was performed at the Ethiopian Public Health Institute (EPHI) and a subculture of all survey strains including ET1291 strain were shipped to the TB Supranational Reference laboratory (SRL) at the San Raffaele Scientific Institute in Milan for further phenotypic tests and whole-genome sequencing. Morphological and growth properties were assessed on Löwenstein–Jensen (BD BBL™, cat. no. 220909), Middlebrook 7H10 (BD Difco™, cat. no. 262710), and MGIT media (BD BBL™, cat. no. 245113). Growth rate in the Middlebrook 7H9 medium (BD Difco™, cat. no. 271310) supplemented with OADC (BD Difco™ BBL™, cat. no. 211886) and 0,5% glycerol (Sigma-Aldrich, Product No. G5516) was monitored using optical density measurements[39]. Extended DST to first- and second-line anti-TB drugs was performed using the BACTEC MGIT 960 System according to the manufacturer's instructions and WHO's recommendations[40]. Minimum inhibitory concentrations (MICs) for bedaquiline, delamanid, and pretomanid were performed with the BACTEC MGIT 960 system[41].

### Targeted next-generation sequencing
Targeted next-generation sequencing analysis was performed on DNA from cultured isolates using the Deeplex Myc-TB kit and the associated web application as per the manufacturer's instructions[22].

### Whole-genome sequencing
**Illumina short-read sequencing, SNP calling, and phylogenetic analysis.** Genomic DNA was extracted and purified using the Maxwell® 16 Cell DNA Purification kit (Promega, USA). Paired-end DNA libraries were prepared using the Nextera XT DNA Library Prep Kit (Illumina, CA, USA) and sequenced on an Illumina NextSeq 500 platform with 150-bp read lengths. The corresponding sequence reads, as well as the Nanopore reads, the associated sequence assembly and annotation (see below), were submitted to the NCBI Sequence Read Archive with Project number PRJNA823537. In order to detect drug resistance-associated mutations and identify strain (sub)lineages, the sequenced raw reads underwent adapter trimming with Trimmomatic[42] and reads shorter than 20 bp were discarded. The resulting reads were aligned to

*M. tuberculosis* H37Rv ATCC 27294 (NC_000962.3) as reference genomes using the MTBseq pipeline[21] by application of the mem algorithm of the Burrows-Wheeler alignment tool v0.7.17[43]. Duplicated reads were marked using the Picard tool v2.23.4-0 (https://github.com/broadinstitute/picard) and local realignment of reads around InDels was performed using the Genome Analysis Toolkit v3.8. SNPs were called with Samtools mpileup v1.6[44] using the following thresholds: minimum mapping quality of 20, minimum base quality at a position of 20, minimum read depth at a position of 8X, maximum strand bias for a position of 90%.

Genome-wide SNP calling of ET1291 and 120 other genomes including 80 representatives of all known human- and animal-adapted lineages of the MTBC (including the recently identified lineage 9[5]), as well as 39 available and one newly sequenced genomes of previously known *M. canettii* strains (Supplementary Data 3), was performed using the CLC Genomics Workbench (version 21.0.5; Qiagen). Reads were mapped on the reconstructed ancestral genome sequence of the MTBC obtained by Comas et al.[45], after masking repetitive regions as defined by self-self BLAST[46], resulting in a final reference sequence of 4,087,113 bp. Reads with more than one match were ignored. Variants were called using the basic variant calling tool with a minimum frequency filter set to 90% (min. coverage 10, min. count 3), and with other parameters set to default. The false positive variant calling rate with this analysis pipeline was verified by mapping Illumina sequence reads from *M. tuberculosis* H37Rv genomic DNA against a previously assembled reference genome obtained after PacBio sequencing of the same genomic DNA preparation, resulting in absence of any detectable SNP. Positions of single nucleotide variants (SNV) identified in the whole dataset were defined using the "Merge Variant Tracks and Identify Known Mutations" tool. A custom script was used to extract 82,637 positions commonly covered in all genomes, to avoid biases due to variable read coverages across the genome dataset, and to analyze recombination and compare inter-strain SNP distances based on a same, normalized number of sequence positions. The utilization of a reconstructed ancestral genome sequence of the MTBC as a common reference for SNP analysis was further justified by previous findings that *M. canettii* and MTBC strains share >89% of their genomes, excluding repetitive regions such as PE_PGRS- and PPE_MPTR-encoding regions (accounting for ~8% of the coding capacity of *M. tuberculosis*), and that individual accessory genomes of *M. canettii* strains show very few genes in common[12].

These SNPs, along with their positions on the reference genome, were used to generate a whole-genome alignment of the same length as the reference genome (4,087,113 bp). This alignment was used as input to build the phylogeny using PhyML v3.3[47] (Fig. 2a). No Lewis correction was necessary since a whole-genome alignment was used as input, rather than an alignment of SNPs. Bootstrap support values were computed using the PhyML default Shimodaira-Hasegawa-like (SH-like) procedure. This phylogeny was then used as the starting point to build a recombination-corrected phylogeny using ClonalFrameML v1.12[25] (Fig. 2b). For each branch, the relative effect of recombination versus mutation (r/m) was computed using the formula $r/m = (d1−d2)/d2$ where d1 and d2 are the lengths of the branches before and after correcting for recombination, respectively. These values of r/m are illustrated by the redness of each branch in Fig. 2b. The genomic locations of the recombination events detected by ClonalFrameML on each branch are shown in Fig. 2c. When SNP differences were computed based on positions covered between genome pairs instead of using positions commonly covered in the entire dataset, the maximum distance between any two MTBC genomes was 2417 SNPs (between *M. orygis* and L7 genomes), and 2096 to 2263 SNPs separated *M. bovis* from *M. tuberculosis* (L1-L4, L7-L8) genomes, in line with previous results[8].

A phylogeny additionally including more distantly related genomes of the MTB-associated phylotype was inferred by using the

Codon Tree pipeline of the Bacterial and Viral Bioinformatics Resource Center (BV-BRC, including tools from the previous PATRIC resources), available at https://www.bv-brc.org/app/PhylogeneticTree. The pipeline uses concatenated nucleotide and encoded amino acid sequences from up to 1000 single-copy core genes, identified via detection of cross-genus BV-BRC global Protein Families (PGFams) homology groups[48], for constructing a RAxML tree. Out of 1160 single-copy core genes identified in all 59 genomes analyzed, 1000 were picked randomly. Corresponding protein and nucleotide coding sequences were aligned using MUSCLE[49] and the Codon_align function of BioPython, respectively. The concatenated alignment of all protein and nucleotide sequences written in a PHYLIP file was partitioned for describing the alignment in terms of protein sequences and first, second, and third codon positions of nucleotide sequences. The resulting file was used for constructing a maximum-likelihood tree using RAxML-NG v1.0.2 with '−model GTR + G + ASC_LEWIS[50]. We used the general time reversible model of nucleotide substitution under the gamma model of rate heterogeneity and performed 1000 alternative runs on distinct starting trees. Support values were generated using 100 rounds of the "Rapid" bootstrapping and the best-scoring maximum-likelihood topology was "midpoint rooted" using FigTree v1.4.4 (https://github.com/cdeanj/figtree). The topology was annotated and colored using the Evolview v3 online tool[51].

**Oxford Nanopore Technologies long-read sequencing and complete genome assembly.** Genomic DNA of ET1291 was purified from culture with the Genomic DNA Clean & Concentrator kit (Zymo Research, Irvine, CA, USA) to recover large-size DNA, according to https://files.zymoresearch.com/protocols/. Input genomic DNA prepared with the Rapid Barcoding Sequencing (SQK-RBK004) kit (ONT, Oxford, UK) was sequenced on a MinION device using a flow cell R9.4.1.

In order to reconstruct a high-quality assembled genome based on the Illumina short reads and ONT long reads, a hybrid genome assembly was constructed as follows. Nanopore long reads were corrected using the Ratatosk de novo error correction tool v0.7.6.3[52] using Illumina paired-end short reads. Corrected long reads were assembled using Flye de novo assembler v2.9.1 (https://github.com/fenderglass/Flye). Initial genome annotation was done using the BV-BRC RASTtk-enabled genome annotation service[53]. A circular map of the chromosome was visualized with BRIG[54]. Comparative alignments and final genome annotation was performed similarly to the strategy described in[4], against an ensemble of genome sequences including representatives of MTBC L1-4 and L7-8 (*M. tuberculosis sensu stricto*; comprising H37Rv), L5-6 &9 (*M. africanum*), *M. bovis*, *M. canettii* strains STB-A, -D, -E, -G, -H, -I, -J and -K, *M. kansasii*, *M. marinum*, *M. lacus*, *M. shinjukuense M. decipiens*, and *M. riyadhense*. These steps were performed based on BLAST searches and analysis of gene synteny, using Artemis and Artemis comparison tool[55] as well as a custom Multiple Annotation of Genomes and Differential Analysis (MAGDA) software previously used for annotation of MTBC L8, *M. canettii* and *Helicobacter pylori* genomes[4,12,56]. Where applicable, annotations were transferred from those of *M. tuberculosis* or *M. canettii* orthologues in the TubercuList/Mycobrowser database or from data from Supply et al.[12], using BLAST matches of >90% protein sequence identity, an alignable region of >80% of the shortest protein length in pairwise comparisons and visual inspection of the gene synteny. Finally, in order to estimate the potential false indel rate in the complete genome assembly, we determined the number of frameshift indels among coding sequences of the orthologues of the 461 genes identified as being essential for in vitro growth of *M. tuberculosis* H37Rv, by comprehensive essentiality analysis[30]. Only three (0.7%) occurrences were detected, indicating a low false indel rate.

Analysis of average nucleotide identity (ANI) of the ET1291 genome versus the genome of *M. tuberculosis* H37Rv ATCC 27294 was done using the ANI calculator online tool with default parameters (http://enve-omics.ce.gatech.edu/).

**In silico spoligotyping, detection, and annotation of CRISPR-Cas loci.** In silico spoligotyping was performed with Illumina sequence reads using the kvarQ software v 0.12.2[57]. To detect clustered regularly interspaced short palindromic repeats (CRISPR) -Cas loci in assembled genomes, the CRISPRCasFinder[58] pipeline (https://github.com/dcouvin/CRISPRCasFinder) was used with default parameters. CRISPR-Cas type and subtype were assigned by using CRISPRmap[59].

### Reporting summary
Further information on research design is available in the Nature Portfolio Reporting Summary linked to this article.

## Data availability
The genome sequencing data of ET1291 generated in this study have been deposited in the NCBI database under Bioproject accession code PRJNA823537, with SRR23035463 and SRR18636082 accession codes for Oxford Nanopore- and Illumina-derived genome sequences, respectively. Source data for Supplementary Fig. 1 are provided with this paper. Source data are provided with this paper.

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

## Acknowledgements

Timothy Walker is acknowledged for sharing the set of repetitive regions in the genome of *M. tuberculosis* H37Rv defined by self-self BLAST. Mayam Omrani, Shahab Saghaie, and Shahram Saghaie are gratefully thanked for their expertise and assistance in Bioinformatics analysis. We thank Alberto Trovato for his assistance in whole-genome sequencing.

## Author contributions

D.C., P.S., A.G., B.Y. and A.M.C. designed the study. A.G. and P.S. analyzed data and wrote the manuscript with comments from all authors. A.G., M.C. and M.GE. conducted phenotypic and molecular characterization of ET1291. A.G. and C.G. performed the assembly of sequences. A.G. and M.M. annotated the ET1291 genome. A.G., C.G. and X.D. performed SNP analyses and phylogenetic reconstruction. X.D. performed recombination analyses. A.G., M.M. and P.S. conducted comparative analyses of complete mycobacterial genomes. B.Y., G.D., E.T., A.M.C., M.A., S.M., A.A., B.D., W.S., H.M., A.M., M.T., D.F.G., Y.A., G.S., B.Z., M.G., G.T., S.A., A.K., M.GE., Z.M. and D.C. ran or contributed to the tuberculosis drug resistance national survey.

## Competing interests

P.S. is a consultant for GenoScreen, which developed the Deeplex Myc-TB kit. C.G. was a GenoScreen employee. The other authors declare no competing interests.
