## [Peer review file · Nature Communications]

A smooth tubercle bacillus from Ethiopia phylogenetically close to the *Mycobacterium tuberculosis* complexEditorial Note:

Reviewer #1 (Remarks to the Author):

This work describes the genome of a tuberculosis clinical isolate from East Africa. The authors find this genome to be in an intermediate position between *M. tuberculosis* and the most closely related *M. canettii* strain. The manuscript shows genomic differences between them, including the presence or absence of genes and SNPs. In my opinion, this is a very relevant finding because it reveals important information about the evolutionary steps of *M. tuberculosis* evolution. Methods are adequate, although I am unsure about some specific analyses on SNP differences and recombination (see comments 5, 6, and 7). The conclusion is supported by the result but some discussion points are not directly addressed in the study (see comment 10). My main concern is about the most relevant data provided, which is the sequencing data, and should be available upon review. However, it is not available to reviewers, which prevents the complete assessment of the study (see comments 2, 3, and 4).

Comment 1. What is the proposed classification of this tuberculosis clinical isolate?

Comment 2. It is not possible to assess the data included in the phylogeny because 40% of the sequences do not have the corresponding accession number in table S3.

Comment 3. Accession numbers for the PacBio and Illumina reads are not public. The sentence "The sequence reads of both strains were submitted to the NCBI sequence read archive with Project number PRJNA823537" on page 14 is not clear. Only the genome of one strain is presented here, so which two strains are referred to here? Moreover, PRJNA823537 is not found in any reads database.

Comment 4. The assembly and annotation are not published or public, and no reference to them is included in the manuscript.

Comment 5. Phylogeny in sup figure 2 is constructed based on 1000 single copy core genes. The explanation at the end of page 15 pointing out a web service is not enough to understand how this was done. Why use this approach instead of generating a core genome and performing a phylogeny? Were the 1000 single-copy genes present in all genomes?

Comment 6. I think the recombination analysis done is not enough to properly assign the recombination rate in each branch of the phylogeny. In my opinion, the results would be more robust with additional analysis using closed genomes and not only the alignment of common SNPs because it can bias the results towards considering preferentially regions in *M. tuberculosis* which is the reference used to call the SNPs. Additionally, Figure 1c shows two *Mtb* branches with a great number of recombination events. I think the authors should comment on this, explaining the impact of this result on the estimate of recombination within *M. tuberculosis*.

Comment 7. The number of SNP differences between ET1291 and *M. tuberculosis* and *M. canettii*, is based on common SNPs called by mapping to an *M. tuberculosis* reference genome. Because ET1291 is in an intermediate position between *M. tuberculosis* and *M. canettii*, I think that the number might be biased towards detecting better the differences with *M. tuberculosis* because common regions in ET1291 and *M. canettii* are not included. Do the authors consider this possible bias? If so, that would support further the similarity of ET1291 and *M. tuberculosis*. One way to estimate it could be to compare the closed genome with closed genomes of *M. tuberculosis* and *M. canettii*.

Comment 8. Figure 1A. I cannot see if there is enough resolution, but, Are animal-associated *M. tuberculosis* lineages a monophyletic group?

Comment 9. Figure 2A and 2B and 2C on page 15 refer to a phylogeny, which is not what the figure shows.

Comment 10. The inference of the date of diversification of the ET1291 that is discussed on page 11 is not directly investigated in the study.

Reviewer #2 (Remarks to the Author):

The authors describe the discovery of a new *M. canettii* isolate from a patient from East Africa. This novel strain is closer to the MTBC than other *M. canettii* strains described to date. They provide phylogenetic and comparative genomic analyses to characterize this isolate. The manuscript is of interest to the TB research community, as it opens a new perspective about MTBC

evolution. The manuscript is overall well written.

I have few concerns, particularly about [redacted].

I will describe this in more detail below, along with other comments.

There were no line numbers in the manuscript, so it was difficult to review it. I tried to indicate the sentences and paragraphs by page numbers. I hope this helps.

Page 4 – LOS means lipooligosaccharides and not liposaccharides

Page 6 – 1st paragraph – please, indicate the type of media in which the growth curves were generated. It would be helpful to also have this information in the legend of the Supp. Fig 1.

Page 7 (1st paragraph) and legend of figure 1 - Please, indicate that the phylogenetic trees were built using a whole-genome alignment, both in the text and figure legend. Also include in figure legend the algorithm used (maximum likelihood).

Figure 1. Please, show or indicate bootstrap values.

Overall, I believe future readers of this manuscript can benefit from more information in the figure legends, as they should be “stand alone” texts. For example, acronyms and bacterial genus can be written in full. Some legends are also not standardized; some have (A) others have A:. They also do not have titles. I have highlighted other points related to them below.

Figure 1C. Can the authors verify why one of the MTBC genomes appeared with recombinogenic regions? Isn't there a problem in the alignment, SNP calling or quality of this genome?

Page 8 (1st paragraph) – normally, the distance between *M. bovis* and *M. tuberculosis* genomes is ~2,000 SNPs. The authors state that the maximum difference between any two genomes of the MTBC was 1,735 SNPs, which is close, but it made me worry a bit. Were there many SNPs among MTBC that were discarded during the recombination adjustment?

Page 8 (2nd paragraph) – in our experience, long read sequencing may result in false indels, even when a hybrid assembly with Illumina reads is performed. This high number of indels results in an overestimation of pseudogenes. Assuming that the assembled and annotated genome will become publicly available, have the authors corrected this problem? PAGP is one of the best tools to detect pseudogenes and once a genome is deposited in GenBank it can be annotated with it. If the genome shows a higher-than-expected number of pseudogenes, it may not be included into RefSeq by the NCBI team.

Table S5 – there is a typo in metabolism.

I could not find the accession number of the assembled genome in the manuscript. The authors only provide a bioproject number, which is not yet available, and mentioned that only the reads were deposited. I suggest to deposit the assembled genome as well to allow reproducibility of the manuscript.

Page 9 (second line) – contains

Table S8 – lines 3 and 4 – “description” is missing.

Page 9 (1st paragraph) and abstract – [redacted]

[redacted]

[redacted]

Page 9 (last paragraph) – in the sentence “ET1291 is phylogenetically distant...” the authors add Figure 4 to show how phylogenetic distant the *M. canettii* strains are; however, figure 4 is related to the CRISPR genes and not phylogeny.

[redacted]

Materials and methods

Page 15 – please, indicate how indels were handled in the generation of this whole-genome alignment.

Page 15 – second paragraph – I guess after PhyML should be figure 1A and not 2A. Figure 2B and 2C are also mentioned in the place of figure 1B and 1C.

Page 15 – second paragraph – were bootstraps run?

Discussion

First paragraph - It would be interesting to date the MRCA of the MTBC and this new isolate to compare to other studies.

Second paragraph – the authors mentioned that the original progenitor emerged from a *M. canettii*-like ancestor at most 6,000-10,000 years ago. This is speculative considering that no dating estimate analysis is provided. The origin of the MTBC MRCA has been a matter of much debate; I believe any speculation on dating estimates from now on should accompany robust analysis. The literature currently carries on with contrasting dating estimates that are propagated erroneously throughout many disciplines, including history. Thus, I believe no conclusion should be drawn without robust analysis. If the authors will not provide dating estimates analysis, I suggest removing this sentence.

Page 12, second paragraph – As mentioned previously, I believe the presented evidence is not strong for these conclusions.

Methods

Page 14, first paragraph – a subculture of all survey strains.... A – a verb is missing in this

sentence.

Middlebrook 7H9 – please, provide the formulation – was it supplemented with OADC? Glycerol?

Page 14 – third paragraph – please, describe if the sequencing was paired-end or single end.
Indicate window size and quality parameter of trimmomatic.

Reviewer #3 (Remarks to the Author):

This study reads well and provides significant insights regarding ancestral *Mycobacterium tuberculosis* strains. The manuscript also indicates interesting results regarding the potential environment of the MTBC progenitor.

However, I have a few comments on this study:

Please add the citation for CRISPRCasFinder (<https://doi.org/10.1093/nar/gky425>).

It would be interesting to add a workflow/pipeline figure showing the different steps you performed for this study (to make it more reproducible).

REVIEWER COMMENTS

Reviewer #1 (Remarks to the Author): This work describes the genome of a tuberculosis clinical isolate from East Africa. The authors find this genome to be in an intermediate position between *M. tuberculosis* and the most closely related *M. canettii* strain. The manuscript shows genomic differences between them, including the presence or absence of genes and SNPs. In my opinion, this is a very relevant finding because it reveals important information about the evolutionary steps of *M. tuberculosis* evolution. Methods are adequate, although I am unsure about some specific analyses on SNP differences and recombination (see comments 5, 6, and 7). The conclusion is supported by the result but some discussion points are not directly addressed in the study (see comment 10). My main concern is about the most relevant data provided, which is the sequencing data, and should be available upon review. However, it is not available to reviewers, which prevents the complete assessment of the study (see comments 2, 3, and 4).

We thank the reviewer for his appreciation on the relevance and the importance of our findings. Regarding the sequencing data, we initially subjected them to an embargo until publication as usually done, but omitted to generate a reviewer link. We apologize for this, and a link to the data is now provided below. The other specific comments have also been carefully addressed, as described hereafter.

Comment 1. What is the proposed classification of this tuberculosis clinical isolate?

This isolate is proposed to be classified as *M. canettii*, as suggested in several places in the results section (e.g. “Targeted next generation sequencing (tNGS) using Deeplex Myc-TB testing resulted in *M. canettii* identification with a predicted pyrazinamide mono-resistant profile, based on *hsp65* sequencing data and detection of specific phylogenetic SNPs including *pncA* A46A”. Like for previous *M. canettii* strains, no spoligotype spacers were detected...”), and in the discussion (e.g. “The discovery in East Africa of a *M. canettii* strain phylogenetically much closer to the MTBC reveals that this was likely not the case...”). We now further clarify this, by adding a new figure (new Figure 1) showing the *M. canettii* identification by the Deeplex Myc-TB results, and by adding the following sentence at the end of the second paragraph of Results: “This *M. canettii* classification was supported by multiple other archetypal characteristics shared with previously described *M. canettii* strains, as described below.”

Comment 2. It is not possible to assess the data included in the phylogeny because 40% of the sequences do not have the corresponding accession number in table S3.

Accession numbers have now been added for all sequences included in this table.

Comment 3. Accession numbers for the pacbio and Illumina reads are not public. The sentence “The sequence reads of both strains were submitted to the NCBI sequence read archive with Project number PRJNA823537” on page 14 is not clear. Only the genome of one strain is presented here, so which two strains are referred to here? Moreover, PRJNA823537 is not found in any reads database. This sentence has been corrected (with removing “of both strains” and indicating Nanopore instead of PacBio reads), as follows: “The corresponding sequence reads, as well as the Nanopore reads, the associated sequence assembly and annotation (see below), were submitted to the NCBI Sequence Read Archive with Project number PRJNA823537”. As indicated above, the Nanopore and Illumina reads will be publicly released upon publication, and are available to the reviewers via this link:

[redacted]

Comment 4. The assembly and annotation are not published or public, and no reference to them is included in the manuscript.

The sequence assembly and annotation are available via the link provided above, and will be publicly available via the BioProject PRJNA823537.

Comment 5. Phylogeny in sup figure 2 is constructed based on 1000 single copy core genes. The explanation at the end of page 15 pointing out a web service is not enough to understand how this was done. Why use this approach instead of generating a core genome and performing a phylogeny? Were the 1000 single-copy genes present in all genomes?

As indicated in Results, this Codon Tree pipeline-based approach was chosen when including additional strains of the MTB-associated phylotype as more external outgroups, for further confirmation of the intermediate phylogenetic position of ET1291 (which was already clearly supported by genome-wide SNP analysis of MTBC and other *M. canettii* strains). Indeed, this approach distinctively uses both protein sequences and corresponding nucleotidic coding sequences for RaxML-based phylogenetic reconstruction. This combined protein/DNA sequence basis thus enables meaningful determination in a single analysis of cross-genus phylogenetic relationships, both among closely related (MTBC and, to a lower degree, *M. canettii*) and with more distantly related (MTB-associated phylotype) genomes. The 1000 single-copy genes that were used were indeed present in all 59 genomes that were analyzed in this comparison.

We now accordingly provide further explanation on this, as follows. “A phylogeny additionally including more distantly related genomes of the MTB-associated phylotype was inferred by using the Codon Tree pipeline of the Bacterial and Viral Bioinformatics Resource Center (BV-BRC, including tools from the previous PATRIC resources), available at <https://www.bv-brc.org/app/PhylogeneticTree>. The pipeline uses concatenated nucleotide and encoded amino acid sequences from up to 1000 single-copy core genes, identified via detection of cross-genus BV-BRC global Protein Families (PGFams) homology groups⁵⁸, for constructing a RAxML tree. Out of 1160 single-copy core genes identified in all 59 genomes analyzed, 1000 were picked randomly. Corresponding protein and nucleotidic coding sequences were aligned using MUSCLE⁵⁹ and the Codon_align function of BioPython, respectively. The concatenated alignment of all protein and nucleotide sequences written in a PHYLIP file was partitioned for describing the alignment in terms of protein sequences and first, second and third codon positions of nucleotide sequences. The resulting file was used for constructing a maximum-likelihood tree using RAxML-NG v. 1.0.2 with ‘-model GTR+G+ASC_LEWIS⁶⁰. We used the general time reversible model of nucleotide substitution under the gamma model of rate heterogeneity and performed 1000 alternative runs on distinct starting trees. Support values were generated using 100 rounds of the “Rapid” bootstrapping and the best-scoring maximum-likelihood topology was “midpoint rooted” using FigTree (<https://github.com/cdeanj/figtree>). The topology was annotated and colored using the Evolview v3 online tool⁶¹”.

Comment 6. I think the recombination analysis done is not enough to properly assign the recombination rate in each branch of the phylogeny. In my opinion, the results would be more robust with additional analysis using closed genomes and not only the alignment of common SNPs because it can bias the results towards considering preferentially regions in *M. tuberculosis* which is the reference used to call the SNPs. Additionally, Figure 1c shows two *Mtb* branches with a great number of recombination events. I think the authors should comment on this, explaining the impact of this result on the estimate of recombination within *M. tuberculosis*.

Thank you for this comment. We think that our recombination analysis is robust and comprehensive for the following reasons, and we provide additional explanations to further clarify this in the manuscript. Using SNPs at positions commonly covered on the ancestral *M. tuberculosis*-based MTBC genome as a same reference for the entire set of strains uniquely allows for analytical comparison of recombination rates on a single, same normalized basis across all branches. Moreover, as indicated in Results, the closed genomes of ET1291 and previous *M. canettii* strains are at most 0.12 Mb larger (for STB-K) than the *M. tuberculosis* H37Rv reference genome, on which the ancestral MTBC genome has previously been reconstructed. Furthermore, in addition to average nucleotide identities of at least 98% with *M. tuberculosis* as indicated in Results, these *M. canettii* strains share the largest part of their genomes with MTBC strains (i.e. >89% for previous closed genomes of *M. canettii* excluding repetitive

regions, as reported in Supply et al., Nature Genetics, 2013; 94% of shared CDSs with *M. tuberculosis* H37Rv in the case of ET1291). Thus, regions not represented in the *M. tuberculosis* reference constitute only a small fraction of the *M. canettii* genomes, which minimizes both the risk and extent of potential bias, if any, in our recombination analysis.

We added a concise explanation in Methods in accordance as follows: “A custom script was used to extract 82,637 positions commonly covered in all genomes to avoid biases due to variable read coverages across the genome dataset, and to analyze recombination and compare inter-strain SNP distances based on a same, normalized number of sequence positions. The utilization of a reconstructed ancestral genome sequence of the MTBC as a common reference for SNP analysis was further justified by previous findings that *M. canettii* and MTBC strains share >89% of their genomes, excluding repetitive regions such as PE_PGRS- and PPE_MPTR-encoding regions (accounting for ~8% of the coding capacity of *M. tuberculosis*), and that individual accessory genomes of *M. canettii* strains show very few genes in common¹².”

With respect to Figure 1c (now Figure 2c), the two branches that are referred to correspond to those leading to the progenitor of the MTBC, and to the ancestor of ET1291 and the MTBC progenitor, respectively. Thus, they are not part of the MTBC and *M. tuberculosis* branches.

This is now further clarified graphically in the figure and the legend as follows: “Note that the red dotted lines indicate two branches that are ancestral to the progenitor of the MTBC and to the progenitor of the MTBC and ET1291, respectively, and are thus not part of the MTBC clade”.

Comment 7. The number of SNP differences between ET1291 and *M. tuberculosis* and *M. canettii*, is based on common SNPs called by mapping to an *M. tuberculosis* reference genome. Because ET1291 is in an intermediate position between *M. tuberculosis* and *M. canettii*, I think that the number might be biased towards detecting better the differences with *M. tuberculosis* because common regions in ET1291 and *M. canettii* are not included. Do the authors consider this possible bias? If so, that would support further the similarity of ET1291 and *M. tuberculosis*. One way to estimate it could be to compare the closed genome with closed genomes of *M. tuberculosis* and *M. canettii*.

As pointed out above, the use of SNPs at positions commonly covered on this same reference is a unique way to compare relative genetic distances on a normalized basis between ET1291, other *M. canettii* and the MTBC genomes, also avoiding thereby biases due to variable read coverages across the dataset. Moreover, common regions of ET1291 and other *M. canettii* strains that are not represented in the *M. tuberculosis*-based reference represent less than one percent of the complete genomes of these strains. Indeed, we anecdotally found only 26 CDSs in common between ET1291 and other closed genomes of *M. canettii*, among the CDSs that are absent from the *M. tuberculosis*-based reference. This is consistent with our previous findings of very few genes shared among individual accessory genomes of *M. canettii* (as defined relatively to MTBC genomes; only nine coding sequences were common to the accessory genomes of all five complete *M. canettii* genomes analyzed in Supply et al., Nature Genetics, 2013). In other words, the use of the reconstructed ancestral MTBC genome as a common reference captures virtually all the variation exploitable for determining SNP-based phylogenetic relationships among any strains considered here, including ET1291, other *M. canettii* and MTBC strains. Therefore, no significant bias is expected. Please see above for the corresponding explanation now included in Methods.

Comment 8. Figure 1A. I cannot see if there is enough resolution, but, Are animal-associated *M. tuberculosis* lineages a monophyletic group?

Consistent with original findings by Coscolla et al. (ref. 5), the animal-associated lineages are found as a paraphyletic group, as out of the 20 animal-associated strains, 6 share a common ancestral node with L6+L9 strains, slightly before the ancestor shared by the remaining 14. However, we believe that this confirmatory detail is beyond the scope of this study.

Comment 9. Figure 2A and 2B and 2C on page 15 refer to a phylogeny, which is not what the figure shows.

Thank you for noticing this, which has been corrected to newly numbered Figure 2A, 2B and 2C.

Comment 10. The inference of the date of diversification of the ET1291 that is discussed on page 11 is not directly investigated in the study.

Please see our response to the related comment by Reviewer 2.

Reviewer #2 (Remarks to the Author): The authors describe the discovery of a new *M. canettii* isolate from a patient from East Africa. This novel strain is closer to the MTBC than other *M. canettii* strains described to date. They provide phylogenetic and comparative genomic analyses to characterize this isolate. The manuscript is of interest to the TB research community, as it opens a new perspective about MTBC evolution. The manuscript is overall well written.

We thank the reviewer for this positive appreciation.

I have few concerns, particularly about [redacted].

I will describe this in more detail below, along with other comments.

There were no line numbers in the manuscript, so it was difficult to review it. I tried to indicate the sentences and paragraphs by page numbers. I hope this helps.

[redacted]

Page 4 – LOS means lipooligosaccharides and not liposaccharides

Thanks for pointing this out. The text was corrected accordingly.

Page 6 – 1st paragraph – please, indicate the type of media in which the growth curves were generated. It would be helpful to also have this information in the legend of the Supp. Fig 1.

As suggested, we now indicate that these growth curves were generated in Middlebrook 7H9 supplemented with OADC and glycerol, in Results and in the legend of Supplementary Fig. 1, in addition to the information previously provided in Methods.

Page 7 (1st paragraph) and legend of figure 1 - Please, indicate that the phylogenetic trees were built using a whole-genome alignment, both in the text and figure legend. Also include in figure legend the algorithm used (maximum likelihood).

Thanks for suggesting this. The text and the figure legend were updated accordingly.

Figure 1. Please, show or indicate bootstrap values. Overall, I believe future readers of this manuscript can benefit from more information in the figure legends, as they should be “stand alone” texts. For example, acronyms and bacterial genus can be written in full. Some legends are also not standardized; some have (A) others have A:. They also do not have titles. I have highlighted other points related to them below.

As suggested, bootstrap values are now indicated for values below 100%; for branches with 100% support, values are not shown, as specified in the updated legend. Legends of this and other figures have been homogenized, and titles and additional information have been added for making them more self-explanatory.

Figure 1C. Can the authors verify why one of the MTBC genomes appeared with recombinogenic regions? Isn't there a problem in the alignment, SNP calling or quality of this genome?

We presume that the reviewer refers here to the branch leading to the common ancestor of the MTBC, thus corresponding to one of the two branches also mentioned in Reviewer 1's comment 6. As indicated in our response to the latter comment, we have now clarified that this branch is not part of the MTBC, thus implying that corresponding recombination events occurred prior to the emergence of the progenitor of the MTBC. As an additional information, the two single tiny dots seen for two (genuine) MTBC branches most likely reflect anecdotal residual noise, linked to analysis of highly genetically conserved bacteria.

Page 8 (1st paragraph) – normally, the distance between *M. bovis* and *M. tuberculosis* genomes is ~2,000 SNPs. The authors state that the maximum difference between any two genomes of the MTBC was 1,735 SNPs, which is close, but it made me worry a bit. Were there many SNPs among MTBC that were discarded during the recombination adjustment?

Thank you for raising this question. The maximum difference between any two MTBC genomes before adjustment for recombination was 1,867 SNPs. The somewhat lower distance, before or after adjustment, relatively to that mentioned by the reviewer rather results from the fact that we imposed a stringent alignment for SNP calculation, by considering only reference positions that were commonly covered in all genomes analyzed, as indicated in Results and Methods. As explained in our responses to Reviewer 1, this is done to avoid biases due to variable read coverages across the genome dataset, and to compare inter-strain SNP distances (i.e. ET1291 vs MTBC and other *M. canettii* strains) based on a same, normalized number of sequence positions. When calculating SNP differences only in a classical pairwise manner, we obtained from 2,096 to 2,263 SNPs between *M. bovis* and *M. tuberculosis* (L1-L4, L7-L8) genomes, and the maximum distance between any two MTBC genomes was 2,417 SNPs (between *M. orygis* and L7 genomes), thus fully in line with similarly calculated pairwise distances in the literature and that mentioned in the comment.

We included this additional information in Methods as follows: "When SNP differences were computed based on positions covered between genome pairs instead of using positions commonly covered in the entire dataset, the maximum distance between any two MTBC genomes was 2,417 SNPs (between *M. orygis* and L7 genomes), and 2,096 to 2,263 SNPs separated *M. bovis* from *M. tuberculosis* (L1-L4, L7-L8) genomes, in line with previous results⁸".

Page 8 (2nd paragraph) – in our experience, long read sequencing may result in false indels, even when a hybrid assembly with Illumina reads is performed. This high number of indels results in an overestimation of pseudogenes. Assuming that the assembled and annotated genome will become publicly available, have the authors corrected this problem? PAGP is one of the best tools to detect pseudogenes and once a genome is deposited in GenBank it can be annotated with it. If the genome shows a higher-than-expected number of pseudogenes, it may not be included into RefSeq by the NCBI team.

We were aware of this point too, based on similar experience with the complete genome assembly of the strain of the new sister clade of the MTBC, combining Illumina and PacBio sequencing data (Ngabonziza et al., Nat. Comm., 2020). As indicated in Methods, we addressed this problem by using the Ratatosk de novo error correction tool - providing substantially increased accuracy of both SNP and indel calls (see Holley et al., Genome Biol, 2021) -, in order to correct the Nanopore long reads using Illumina paired-end short reads. We subsequently assembled the corrected reads by using Flye de novo assembler, which has also been shown to generate more accurate complete assemblies than other assemblers (Kolmogorov et al., Nat. Biotech., 2019).

Moreover, in order to estimate the potential false indel rate in the complete genome assembly, we determined the number of frameshift indels among coding sequences of the orthologues of the 461 genes that were identified as being essential for *in vitro* growth of *M. tuberculosis* H37Rv, by comprehensive essentiality analysis (DeJesus et al, mBio, 2017). The culminating quality of assembly

and finishing of the *M. tuberculosis* H37Rv is widely recognized, and the catalog of 461 essential genes established via completely saturating transposon mutagenesis by DeJesus et al. is considered as authoritative. Therefore, we think that the combined use of this specific high-quality H37Rv genome reference and these comprehensive reference data on gene functionality provides a reliable and more direct estimate of false pseudogenes than when using PGAP. By using this approach, only three (0.7%) occurrences were detected, indicating a low false indel rate.

The description of this verification has been added in Methods, as follows: "Finally, in order to estimate the potential false indel rate in the complete genome assembly, we determined the number of frameshift indels among coding sequences of the orthologues of the 461 genes identified as being essential for in vitro growth of *M. tuberculosis* H37Rv, by comprehensive essentiality analysis⁶⁷. Only three (0.7%) occurrences were detected, indicating a low false indel rate".

Table S5 – there is a typo in metabolism.

Thanks for spotting the typo. The text was amended accordingly.

I could not find the accession number of the assembled genome in the manuscript. The authors only provide a bioproject number, which is not yet available, and mentioned that only the reads were deposited. I suggest to deposit the assembled genome as well to allow reproducibility of the manuscript.

As mentioned above, the sequence data has been deposited to GenBank, along with the sequence assembly and annotation, and are planned to be made publicly released upon publication. These data are available to the reviewers via the following link: [redacted]

Page 9 (second line) – contains

Thanks for this. The text was amended accordingly.

Table S8 – lines 3 and 4 – "description" is missing.

There is no specific description available for these two sequence hits from *Mycobacterium riyadhense*, as per the NCBI BLAST results. This is now indicated as such in Table S8.

[redacted]

[redacted]

Page 9 (last paragraph) – in the sentence “ET1291 is phylogenetically distant...” the authors add Figure 4 to show how phylogenetic distant the *M. canettii* strains are; however, figure 4 is related to the CRISPR genes and not phylogeny.

Thanks for mentioning this. This information is indeed shown in Supplementary Figure 5. The text was updated accordingly.

[redacted]

[redacted]

[redacted]

Materials and methods Page 15 – please, indicate how indels were handled in the generation of this whole-genome alignment.

Only SNPs were used for constructing the whole genome alignment and subsequent phylogenetic reconstruction, as canonically done. Please see our response to the related comment referring to Page 8 (2nd paragraph), for the verification of indels on the complete genome assembly.

Page 15 – second paragraph – I guess after PhyML should be figure 1A and not 2A. Figure 2B and 2C are also mentioned in the place of figure 1B and 1C.

Thanks for mentioning this. The text was updated accordingly, accounting also for the new Figure 1..

Page 15 – second paragraph – were bootstraps run?

Yes. We now indicate that as follows: “Bootstrap support values were computed using the PhyML default Shimodaira-Hasegawa-like (SH-like) procedure.”

Discussion First paragraph - It would be interesting to date the MRCA of the MTBC and this new isolate to compare to other studies.

Please see our response to the next comment, related to the same question.

Second paragraph – the authors mentioned that the original progenitor emerged from a *M. canettii*-like ancestor at most 6,000-10,000 years ago. This is speculative considering that no dating estimate analysis is provided. The origin of the MTBC MRCA has been a matter of much debate; I believe any speculation on dating estimates from now on should accompany robust analysis. The literature currently carries on with contrasting dating estimates that are propagated erroneously throughout many disciplines, including history. Thus, I believe no conclusion should be drawn without robust analysis. If the authors will not provide dating estimates analysis, I suggest removing this sentence. Our study dataset includes representative genomes of all known MTBC lineages (implying selection for diversity) in addition to all available *M. canettii* genomes, with a sample size and distribution of sampling times that do not bode well for an independent dating analysis. Therefore, we used a previously hypothesized date range for the MRCA of the MTBC for suggesting an estimate based on the comparison of clonal distances between ET1291 and the MTBC and the maximal distance between extant MTBC strains. This hypothesized date range (and its underlying molecular clock) is probably the best available to date, as it was obtained by a systematic study done by Gagneux’s group of the molecular clock of MTB on a large genome data set (6,285 strains), covering different epidemiological settings and across the main strain lineages (ref. 37). Moreover, this date range confirmed the range obtained in a landmark study, using a molecular clock calibrated on analysis of archeological DNA (Bos et al. Nature, 2014; ref. 34).

Nevertheless, we agree that even this best estimate remains debated. Therefore, in accordance with the reviewer’s comment, we acknowledge this limitation, we removed specific dating estimates, and we limit ourselves to cautiously consider a possible relative evolutionary scale under a hypothesis of molecular clock conservation, as follows: “By hypothesizing similar molecular clocks across the respective evolutionary branches, this inferred phylogenetic distance would thus imply that the original MTBC progenitor that emerged from a *M. canettii*-like ancestor in East Africa is perhaps only twice as old evolutionarily than the currently reconstructed MRCA of the MTBC. More data would be required for estimating a dating for this emergence, also in the view of the debated dating of the MTBC MRCA^{34,37}”.

Along the same lines, we secondarily removed absolute dating in the first paragraph, which now reads: “Whether lineages of MTBC strains, now extinct, could have existed and caused TB in humans and animals much earlier (maybe by two orders of time magnitude) than the MRCA of known extant lineages, is an open and largely debated question^{10,36,37}”.

Page 12, second paragraph – As mentioned previously, I believe the presented evidence is not strong for these conclusions.

[redacted]

Methods Page 14, first paragraph – a subculture of all survey strains.... A – a verb is missing in this sentence. Middlebrook 7H9 – please, provide the formulation – was it supplemented with OADC? Glycerol?

Thanks for spotting this. The text was amended accordingly, including the information that 7H9 was supplemented with OADC and 0.5 % glycerol.

Page 14 – third paragraph – please, describe if the sequencing was paired-end or single end. Indicate window size and quality parameter of trimmomatic.

Paired-end sequencing was used, as described in the following sentence: “Paired end DNA libraries were prepared using the Nextera XT DNA Library Prep Kit...”. We only used Trimmomatic to remove the adapters and to discard the reads <20 bp, without using the “SLIDING WINDOW” part. This is now clarified in Methods as follows: “In order to detect drug resistance associated mutations and identify strain (sub)lineages, the sequenced raw reads underwent adapter trimming with Trimmomatic⁵² and reads shorter than 20 bp were discarded.”

Reviewer #3 (Remarks to the Author): This study reads well and provides significant insights regarding ancestral Mycobacterium tuberculosis strains. The manuscript also indicates interesting results regarding the potential environment of the MTBC progenitor.

Thank you for your positive comments on the significance and interest of our findings.

However, I have a few comments on this study:

Please add the citation for CRISPRCasFinder (<https://doi.org/10.1093/nar/gky425>).
The reference was added accordingly.

It would be interesting to add a workflow/pipeline figure showing the different steps you performed for this study (to make it more reproducible).

Thanks for this comment. A workflow of this study was added as a new Figure 1 (also showing the results of targeted next sequencing that identified ET1291 as *M. canettii*).

Reviewer #1 (Remarks to the Author):

The authors have explained and addressed my concerns, however, due to the new data provided I have two further comments:

1-The manuscript and response to reviewers states "This isolate is proposed to be classified as *M. canettii*. However, the annotation provided as uploaded to ncbi shows: DEFINITION *Mycobacterium tuberculosis* isolate Clinical isolate chromosome", and "/organism="Mycobacterium tuberculosis" . I guess this has to do with the sequence similarity with the MTBC. I think this discrepancy in classification between the manuscript and the submitted sequence should be addressed. In that sense, the authors report the average nucleotide identity versus *M. tuberculosis* H37Rv of the ET1291 genome is 99,55%. To support further the similarity to MTBC, it would be illustrative to compare it with the average nucleotide identity against the closer *M. canettii* strains.

2- Regarding annotation: I think it would be important to clarify what is the difference between the different descriptions of loci such as "Highly similar to", "Similar to" and "Equivalent to" H37Rv numbers or other genomes. If they correspond to different degree of similarity, it would be informative to indicate the thresholds used.

3- Another question regarding the annotation, I wonder why there are no loci annotated as pseudogenes. I can see there are loci described as "similar to pseudogenes", but no pseudogene annotation is directly found.

Reviewer #2 (Remarks to the Author):

The authors have provided responses to most of the concerns raised, and the manuscript has improved substantially. However, I am still not comfortable with their conclusion about [redacted].

[redacted]

[redacted]

[redacted]

L111 – Is reference 20 the correct one for this sentence?

L202 – 70-90% identity at the nucleotide or amino acid level? Please, indicate.

REVIEWERS' COMMENTS

Reviewer #1 (Remarks to the Author):

The authors have explained and addressed my concerns, however, due to the new data provided I have two further comments:

1-The manuscript and response to reviewers states "This isolate is proposed to be classified as *M. canettii*. However, the annotation provided as uploaded to ncbi shows: DEFINITION Mycobacterium tuberculosis isolate Clinical isolate chromosome", and "/organism="Mycobacterium tuberculosis" . I guess this has to do with the sequence similarity with the MTBC. I think this discrepancy in classification between the manuscript and the submitted sequence should be addressed. In that sense, the authors report the average nucleotide identity versus *M. tuberculosis* H37Rv of the ET1291 genome is 99,55%. To support further the similarity to MTBC, it would be illustrative to compare it with the average nucleotide identity against the closer *M. canettii* strains.

The *M. tuberculosis* identifiers and organism name for the BioSample were mistakenly implemented by the NCBI. This is thus unrelated with the sequence similarity with the MTBC, and has been rectified in the meantime into "*Mycobacterium canettii*". To note, all *M. canettii* strains are categorized as *M. tuberculosis* complex by NCBI Taxonomy, as is the case for ET1291. However, as per its own disclaimer, the NCBI states well that this is not an authoritative source for nomenclature or classification, and invites to consult the relevant scientific literature for the most reliable information. We described elsewhere the multiple biological features that dichotomically distinguish *M. canettii* from the MTBC, e.g. in Supply et al., Nature Genetics, 2013 and Boritsch et al., Nature Microbiology, 2016.

Nevertheless, to avoid entering further into a taxonomic debate, we chose to only use ET1291 for referring to the strain in the manuscript.

2- Regarding annotation: I think it would be important to clarify what is the difference between the different descriptions of loci such as "Highly similar to", "Similar to" and "Equivalent to" H37Rv numbers or other genomes. If they correspond to different degree of similarity, it would be informative to indicate the thresholds used.

Thank you for mentioning this. Indeed, this corresponds to different degrees of similarity. We could not include individual scores in annotation notes due to NCBI rules. In the legend of the corresponding supplementary table, we added that "Highly similar" corresponds to > 95% identity at amino acid level, "Weakly similar" to 50% identity, "Similar" to identity between 50 and 95%, and we replaced "Equivalent" by "Highly similar" for the sake of simplicity.

3- Another question regarding the annotation, I wonder why there are no loci annotated as pseudogenes. I can see there are loci described as "similar to pseudogenes", but no pseudogene annotation is directly found.

As can be seen in multiple examples in "Note" column in Supplemental Data 4 (former Table S4), such loci are annotated more specifically, by referring to the (suspected) mutational event involved, e.g. "disrupted by an in frame stop codon", "Believed to be disrupted by a frameshift mutation". More generic annotations, i.e. "Equivalent to pseudogene *Rv3128c* (193 aa) from *M. tuberculosis* strain H37Rv" and "Probable pseudogene", were used for MUW33_3272 and MUW33_3368, respectively.

Reviewer #2 (Remarks to the Author):

[redacted]

L111 – Is reference 20 the correct one for this sentence?

Yes, it is.

L202 – 70-90% identity at the nucleotide or amino acid level? Please, indicate.

Thank you for asking. We now added that this is at nucleotide level.